# Patient’s Perspective of Telemedicine in Poland—A Two-Year Pandemic Picture

**DOI:** 10.3390/ijerph20010115

**Published:** 2022-12-22

**Authors:** Sebastian Sołomacha, Paweł Sowa, Łukasz Kiszkiel, Piotr Paweł Laskowski, Maciej Alimowski, Łukasz Szczerbiński, Andrzej Szpak, Anna Moniuszko-Malinowska, Karol Kamiński

**Affiliations:** 1Department of Population Medicine and Lifestyle Diseases Prevention, Medical University of Bialystok, Waszyngtona 13A, 15-089 Białystok, Poland; 2Doctoral School, Medical University of Bialystok, 15-089 Bialystok, Poland; 3Society and Cognition Unit, University of Bialystok, 15-403 Bialystok, Poland; 4Doctoral School of Social Sciences, University of Bialystok, 15-403 Bialystok, Poland; 5Department of Endocrinology, Diabetology and Internal Medicine, Medical University of Bialystok, 15-276 Bialystok, Poland; 6Clinical Research Centre, Medical University of Bialystok, 15-276 Białystok, Poland; 7Witold Chodźko Institute of Rural Medicine, 20-090 Lublin, Poland; 8Department of Infectious Diseases and Neuroinfections, Medical University of Bialystok, 15-089 Białystok, Poland; 9Department of Cardiology, University Hospital of Bialystok, 15-276 Białystok, Poland

**Keywords:** telemedicine, telehealth, patient’s perception, COVID-19, SARS-CoV-2, implementation barriers

## Abstract

The outbreak of the COVID-19 pandemic caused the healthcare system to drastically reduce in-person visits and suddenly switch to telemedicine services to provide clinical care to patients. The implementation of teleconsultation in medical facilities was a novelty for most Polish patients. In Poland, the main telehealth services were provided in the form of telephone consultations. The aim of this study is to determine patients’ perceptions of telemedicine in the context of their experiences with the healthcare system during the COVID-19 pandemic. In this study, we presented how the evaluation of telemedicine services from the perspective of patients in Poland changed in the context of the ongoing pandemic. We conducted two surveys (year by year) on a representative quota sample of the Polish population (N = 623). This ensured that our observations took into account the evolution of views on telemedicine over time. We confirmed the well-known relationship that innovations introduced in the healthcare sector require a longer period of adaptation. We also identified significant concerns that limit the positive perception of telemedicine and compared them with experiences described in other countries.

## 1. Introduction

The process of digitization, present in most areas of the economy, has also entered the health sector. Patients are searching for health information online, trying to solve simple and sometimes more complex health problems based on the information available on websites [1,2]. A natural step in opening up to the digital world is the potential willingness to use elements of e-health, including telemedicine. The COVID-19 pandemic has affected various aspects of health systems and requires extensive planning in terms of organising provision, availability of human resources and ensuring the sustainability of the health system. The course and severity of the pandemic has prompted governments in many countries to adopt intensive protocols, such as lockdown strategies, to limit new infections and reduce the societal burden of disease and mortality [3]. In Poland, the first changes to the delivery of medical care in health facilities were introduced at the beginning of March 2020. Lockdown caused by the ongoing COVID-19 pandemic has boosted the adoption of digital health technologies such as teleconsultation in primary health care (PHC) [4].

The first months of the pandemic proved overly challenging for the health system. As the pandemic escalated, the care of patients infected with COVID-19 became a top priority. Face-to-face medical consultations were used almost exclusively in emergencies to minimize the risk of doctors and patients contracting a new disease. In this difficult situation, telemedicine enabled patients to continue their medical care, especially for minor ailments and illnesses [5].

The health care system in Poland is based on common health insurance. Nearly 70% of expenditure is financed by public health insurance contributions. All insured persons have access to a wide range of health services, including primary health services as well as specialized medical services such as outpatient care or hospital care. The provision of health services and their financing are clearly separated: The National Health Fund (NHF), the main funder in the system, is responsible for financing health care and contracting with public and non-public providers. The Ministry of Health has a crucial role in setting health policy and has administrative responsibility only for those health care facilities that it funds directly [6]. Primary health care (PHC) is the first level of professional medical care that enables patients to meet their health needs. The aim of primary health care in Poland is to provide comprehensive and coordinated health care services at the place of residence to all eligible persons [7]. The COVID-19 pandemic caused a significant increase in the use of e-health applications. Soon after the first case of COVID-19 was confirmed in Poland, the Ministry of Health mandated the widespread use of remote medical visits as a substitute for traditional visits to the doctor’s office. It was estimated that, in the first year of the pandemic, as many as 80% of primary care visits were conducted remotely, mainly by telephone [8].

Telemedicine is the use of medical information transmitted via electronic communication (including applications and services using two-way video, email, smartphones, wireless tools and other telecommunications technology) to improve a patient’s clinical condition [9]. In Poland, telemedicine is a relatively new healthcare delivery tool. Legally providing healthcare services remotely started to be possible at the end of 2015, when the amendment of the Health Information System Act introduced the possibility of examining a patient via information communications technology systems (ICT). E-visits became an equally valid health care service. However, this legislative change did not have a significant impact on the functioning of the entire healthcare system. In fact, online visits were introduced into the system within PHC on 5 November 2019 by decree of the Minister of Health [10]. Due to the SARS-CoV-2 pandemic, teleconsultations could be reimbursed in primary care from August 2020. Telemedicine has many benefits, as it enables efficient service delivery and improves communication between medical staff and patients, reducing the necessary time to provide services [11]. Remote consultation is a tool of telemedicine and has been defined as a medical consultation via telecommunications at a distance, usually for the diagnosis or treatment of a patient at a location far from the patient or general practitioner (GP). A teleconsultation is defined as advice in the form of a telephone call, a video call using communicators and a video call via various platforms. Properly carried out, teleconsultation brings significant value in terms of rapid access to medical information, improved quality of healthcare and increased trust in medical staff [12]. Due to the lack of physical presence of the patient, the final diagnostic decision is mainly based on the information provided by the patient to the doctor. The quality of the decision depends primarily on the quantity and quality of the information received or the electronic medical records available. It is also important that it is possible to provide full documentation: medical documents in the form of photographs or video recordings [13].

The aim of this study is to identify patients’ perceptions of telemedicine in the context of their experience with the healthcare system during the COVID-19 pandemic.

## 2. Materials and Methods

The data used in this paper come from the project Rise or fall? Short and long-term health and psychosocial trajectories of COVID-19 pandemics (ROF) authorized by the Polish National Science Centre. A survey questionnaire using the CAWI technique with questionnaires (40 min) was carried out in March 2021 on a representative sample of N = 1000 and was repeated in April 2022 mainly on the same sample (N = 623 + 377 new respondents additionally drawn; Figure 1). The postponement was due to the war in Ukraine, which started in February, translating into considerable social unrest in Poland. The sample was balanced in terms of national quotas for age, gender, provinces (regions), class of residence and level of education. Respondents were drawn from Poland’s largest opinion panel service (opinie.pl, with over 125,000 active panelists), whose provider IQS Group is affiliated with OFBOR (Polish Association of Public Opinion and Market Research Firms) and holds ISO and PKJPA (Programme for Quality Control of Interviewers’ Work) certificates for the CAWI technique. Thanks to this technique, we were able to fully control the survey through filters and transitions between questions and eliminate the risk of missing data. The survey was carried out by the IQS Group. The draw was made until the quotas were filled in, then the results were matched to the structure of the general population of Poles. In total, 623 people were covered with both measurements and these were the sample for the analysis in this paper. Both surveys were carried out in accordance with the highest methodological standards, after the research tools were approved by the Bioethics Committee of the Medical University of Białystok. The questionnaire was structured in blocks to reflect the respondent’s individual circumstances before and during the pandemic: metrics, subjective assessment of health and fitness; experience of SARS-CoV2 and COVID-19; in-depth health interview; pain perception; diet; social life and religiosity; assessment of institutional and government activities; social distance, hygiene and disinfection; vaccination; professional work; stress perception; use of stimulants; family and financial situation; attitudes towards COVID-19; health literacy; assessment of the health system and strategy towards the pandemic. The questionnaire was modelled on existing tools, i.e., the European Social Survey (https://www.europeansocialsurvey.org/, accessed on 10 January 2022), questions from The Joint Research Center COVID-19 Survey (European Commission’s science and knowledge centre-https://ec.europa.eu/eusurvey/runner/JRC-Covid19-Survey, accessed on 10 January 2022), International Social Survey Program (ISSP-Health module, http://www.issp.org/menu-top/home/, accessed on 10 January 2022). The survey also included questions identical to the interview used in the Bialystok PLUS cohort study conducted by the Medical University of Bialystok.

The fact that telemedicine services were used during the study period was prepared on the basis of two survey questions (During the COVID-19 pandemic, did you have a tele-consultation or an e-visit? At a GP/At a specialist), which were scaled to 0/1 to then categorize patients in terms of frequency of telemedicine use after the first and second years of the study period (variables Telemedicine use in 2021, Frequency of telemedicine use during the 2 years of pandemic). A dedicated survey question was used to assess opinions on telemedicine, with respondents selecting a value from a scale of 0 to 10 (totally disagree to strongly agree) in relation to established statements. These were in particular: ‘They are inefficient’; ‘They save my time: e.g., I can easily get a prescription, get a referral’; ‘It happened that the doctor was not able to see the results of my examinations’; ‘I felt unprotected’; ‘I am afraid that through tele-counselling or e-visit a serious illness will not be detected in time for me or my relatives’; ‘Because of the COVID-19 pandemic, I consider the tele-counselling and e-visit system to be appropriate’; ‘It is very difficult to get tele-counselling or e-visit when I or my child need it most’. Items needed to have the scale reversed so that it was uniform with the others and positive in nature (higher value = positive rating). In order to get a clear picture and overall direction of the evaluation of telemedicine services, the average score of all 7 items in each year was calculated. Tables presenting the scale were included in the Appendix A. These questions were checked for reliability and internal consistency with Cronbach’s alpha test with α values of 0.78 in 2021 and 0.85 in 2022.

### Statistical Analysis

All variables used were checked for normality of distribution using the Shapiro-Wilk test. Due to the lack of normal distributions, it was considered reasonable to use non-parametric tests in hypothesis testing. Cross-validation of nominal variables between patient groups was performed using Pearson’s χ^2^ test and opinion values between groups using the Kruskal-Wallis test. Pairwise comparisons were performed using the Mann-Whitney test method, taking into account the Bonferroni correction for multiple comparisons. Data were visualized using radar and box plots. Univariate and multivariate linear regression modeling with backward elimination of variables was carried out to test the association of overall telemedicine service rating (mean), and the final model is presented in a table. Statistical hypotheses were verified at a significance level of *p* = 0.05.

## 3. Results

A group of 623 people was included in the analysis from a nationally representative quota sample (N = 1000) established in March 2021, which was re-surveyed approximately one year after the first contact. The characteristics of the group are presented in Table 1; the frequency of telemedicine use was increased with higher age of respondents, size of residence and level of education. In order to analyze patients’ opinions on telemedicine, they were divided into categories of frequency of use of these services. In addition, opinions on telemedicine were assessed after the first and second year of the SARS-CoV-2 pandemic, taking into account the change during the period under study. Those who did not use telemedicine during the study period answered the same questions as the others, determining how much they agreed with the statement.

Evaluation of the use of telemedicine services during the pandemic period was carried out using a 10-item scale on 7 items—opinions on telemedicine services. After the first year of the pandemic, respondents were divided into categories reflecting the frequency of telemedicine use (never, once, twice). Opinion on time-saving through this type of service had the highest agreement rating regardless of the frequency of use of telemedicine; on the other hand, in all groups, the greatest concern was the risk of not detecting a serious illness without personal contact during the examination (Figure 2). There were clear, significant differences in the evaluation of telemedicine according to the frequency of its use—it was better evaluated when it was used more frequently. Table 2 shows the *p*-value of the tests comparing the frequency distributions of the ratings assigned by respondents to each item. The aggregated average score by the group is presented in Figure 3.

In the study group, an identical analysis was conducted almost exactly one year after the first one. This means that respondents at that stage had two years of COVID-19 pandemic experience behind them. As before, item analysis was conducted in groups reflecting the frequency of telemedicine use over two years. The combination of the two years of observations as a whole, i.e., a summary of the frequency of use of telemedicine services, is shown in Figure 4. Respondents were cataloged according to the categories: never, rarely (1 time in two years) and regular (used telemedicine at least twice in two years). Larger areas of the figures plotted on the radar chart indicate an improving overall assessment of telemedicine in Poland. Responses indicating doctor’s access to medical records during a teleconsultation, accessibility of making an appointment, or general assessment of telemedicine as the right solution for a pandemic may mean that organizational and functional aspects of telemedicine services have improved at the scale of the whole health system (Figure 4). This allowed us to confirm that each of the elements influencing the overall value of telemedicine evaluation was rated significantly higher in the group of regular users (Table 3, Figure 5).

Figure 6 shows the difference between the mean values of the individual items (rated on a scale of 0–10) forming the overall opinion on telemedicine in the second measurement year (2022) and the first (2021). In all assessed items, the value of the difference was positive, so the overall direction of change in attitude towards telemedicine is positive.

Univariate and multivariate regression models confirmed a significant association between age, frequency of telemedicine use and the fact of being vaccinated against COVID-19 and patients’ average rating of telemedicine. Overall, those who used the remote form of medical care more frequently and those who were older rated this type of healthcare delivery better. It is also interesting to note that those who were vaccinated rated telemedicine less favorably, which may mean that they would expect to be able to have traditional contact with a doctor in view of minimizing the risk of virus infection (Table 4).

## 4. Discussion

The COVID-19 pandemic has dynamized the use and adoption of telemedicine by health systems [14]. Telemedicine is an efficient communication tool between providers and patients that prevents the risk of exposure to infected individuals. In addition, it can enhance access to healthcare and improve health outcomes in underserved communities by eliminating structural barriers such as transport, excessive waiting times, inconvenient appointments and regional shortages of healthcare professionals [15]. The success of telemedicine implementation significantly depends on the attitudes of health system stakeholders. Barriers to the implementation of telemedicine most often include the individual level (i.e., low digital abilities, preconceptions and fear of the unfamiliar) as well as the structural level (i.e., geographical location, network access, possession of necessary equipment) [16,17]. Although telemedicine in Poland is relatively new, it has been widely used in other countries such as the United States, Canada and Australia from much earlier in time [18]. Therefore, in our study we aimed to identify patients’ opinion on telemedicine during the COVID-19 pandemic.

The results of our survey reveal that 21.8% of respondents have never used telemedicine during a pandemic. A similar issue was studied by the National Health Insurance Fund in collaboration with the Ministry of Health. They conducted a survey of patient satisfaction with teleconsultations with primary care physicians during the COVID-19 epidemic, showing that 80% of medical consultations took place via telephone. Furthermore, the results showed that teleconsultations were positively perceived and recognized by the majority of patients who used them as a necessary and valuable channel of contact with their doctor [19]. The agency Procontent Communication, on the other hand, drew different conclusions, in a report conducted in January 2022. From the agency’s findings, we find out that more than half of Poles have a negative opinion of the functioning of the health service during the pandemic. Among the main complaints were the poor quality of medical advice (26.4%), the prolonged waiting time for diagnosis (33%) and GP consultation (31%) [20]. We believe that the high percentage of people who did not use telemedicine may be due to the aforementioned problems with accessibility to doctors, which include, for example, staff shortages and doctors being overloaded with work during the pandemic, less frequent personal contact with the patient, as well as staff being delegated to fight COVID-19 and the temporary reduction of specialist advice in many medical facilities. In addition, it should be mentioned that for most Polish patients the teleconsultation system implemented after the outbreak of the pandemic was a novelty. Hence, this could have led to a perception of disorientation. As a result, many patients were obliged to adapt to the new reality of the healthcare system, which was completely unfamiliar to them [21].

Danish patients also found themselves in a similar situation. In Denmark, the telemedicine consultation system was first introduced in March 2020, in response to the increasing number of new cases of COVID-19 infection. At that time, the Danish Health Authority ordered that face-to-face visits needed to be replaced by telephone consultations to reduce the risk of transmission of the virus among patients. Nevertheless, according to a patient satisfaction survey, patients reported a high overall satisfaction rate with telemedicine/telephone consultations (N = 238; 85.0%) throughout the global pandemic. Thus, the percentage was significantly higher compared to our study. Moreover, these patients were not very interested in upgrading their technological approach to include video consultations. As the authors noted, age, gender and distance from the hospital were also unrelated to satisfaction with the telephone consultation. Among patients, the main reason for declining the proposal to integrate video consultations into the health system was the lack of necessary equipment (55.1%). Numerous respondents also indicated (34.2%) that they would refuse to use telemedicine with video because they would not feel comfortable seeing themselves on camera [22].

In the United States, telemedicine has been used in clinical settings for several decades [23]. Years of experience and research conducted have allowed researchers in Northern California to categorize the most common barriers to the use of video visits. They identified barriers on an individual level, comprising older age and limited digital literacy [17]. Patients with these characteristics avoided or had a negative perception of video visits, had limited access to devices, had more difficulty using video technology, and required assistance. In fact, for this group, the use of audio-only telemedicine services (telephone consultation) appeared to be an important facilitating factor. This factor was confirmed in a study conducted in Los Angeles. In addition to the aforementioned technological inequalities, the authors noted that, for some of the population, another barrier was the fear of losing the relationship with the GP [24]. This data was related to minority populations, e.g., “familismo”, who value personal contact with the doctor [25]. Another barrier with a strong impact on the implementation of telemedicine services was an awareness barrier [26]. In these cases, patient resistance to adopting non-traditional models was a frequently observed problem. Consequently, patients may have had concerns about whether a remote medical examination would be as reliable and as effective as one performed in a doctor’s office [27]. However, this barrier mainly affects older people and those from backgrounds with limited technological development and access to telemedicine systems [26]. Indeed, our results confirm that Polish patients fear that, due to teleconsultations, some serious conditions may not be detected in time. This is a worrying finding, suggesting that the quality of teleconsultations provided in Poland may be rather low. According to a report by the Patient Ombudsman, in January–September 2020 the prevailing opinions of patients concerned access to teleconsultation and emerging problems with its implementation [28]. The list of impediments in primary health care (PHC) was rather long and consisted of conditions such as struggling to reach the patient’s registration, objections towards the quality of teleconsultation, or doubts about the effectiveness of teleconsultation diagnostics. Over time, the widespread use of telemedicine has increased patients’ awareness of how they can reach their PCP in the current situation. The results of our study show that, over the two years of the pandemic, patients recognized the positive time-saving aspect of telemedicine. Furthermore, we found that satisfaction in this context increased significantly in the second year of the pandemic. The observation that positive perceptions of telemedicine increase with frequency of use has been reported in several studies [29,30,31].

Therefore, it is important to reflect on the reasons why, in some countries, the implementation of telemedicine, despite great efforts, meets with stakeholder resistance. We may find an answer to this in the work of Ah Young Kim et al. [32]. The authors described the unusual case of Korea, with telemedicine, with its new technology base, still not having gained popularity 32 years after the first pilot project. The very reason for resistance, as they note, is merely the introduction of a new system into an existing one. The researchers highlighted that telemedicine has not yet been introduced in many countries due to its low acceptance among users, which is due to social, technical, political and legal factors. Among these, social factors are seen as the most problematic [33]. A comprehensive overview compiled by researchers at the Technical University of Dresden provides more information on the barriers to telemedicine implementation Through a detailed analysis of international studies related to the implementation of telemedicine initiatives into regular care, they demonstrated that the introduction of telemedicine is hindered worldwide, regardless of the political system, legal framework or development status [34]. Indeed, as described earlier, particularly, the individual characteristics and skills of patients and healthcare providers as well as sufficient regional infrastructure are the most relevant identified categories of barriers [35].

Finally, one study assessed patient perspectives on telehealth and factors associated with higher satisfaction. Different from our study, they found greater telehealth satisfaction among younger and female patients [36]. However, our observations did not show statistical significance in relation to gender and age. Our study showed that those who regularly used telemedicine services rated them significantly positively compared to those who did not. In the study cited above, there was no correlation between frequency of teleconsultation use and satisfaction. We presume that the negative evaluation of telemedicine by people who have not used it before may be due to the individual concerns of the respondents. As described in previous work, patients’ lack of knowledge, unfamiliarity with communication technologies and fear of the unknown are well-known reasons for the lack of acceptance of telemedicine [37,38]. Therefore, we believe that educating and making patients aware of the impact that telemedicine can have on access to quality healthcare, overcoming not only geographical barriers but also socio-economic barriers, is very important. This may help to reduce patients’ anxiety and improve their experience of telemedicine visits [39].

In summary, it should be emphasized that our study is one of many attempting to analyze patients’ perceptions of telemedicine. Due to the design of our study, we were able to check the changing opinions on telemedicine in the same population year after year, which is rare from a literature perspective. It should also be noted that the rapid, compulsory implementation of telemedicine results in somewhat different perceptions of this type of service than when it runs on a planned, staggered basis over many years. The studies from the USA or Germany cited above point to different factors favoring implementation (such as young age), whereas in our study such a factor was not relevant due to the lack of any other option than telemedicine. The decisive factor was, therefore, health need. What is noticeable in our study and in the Polish health system is that patients, having no other option, used telemedicine care, but the main concerns are the risk of not detecting a serious disease in time, as well as low diagnostic and therapeutic efficiency. The assessment expressed by the Dresden research team that opinions on the implementation of e-health solutions should be studied taking into account the specifics of the country’s political-administrative system, cultural and demographic conditions of the country in question is therefore correct. It seems that, as the positive evaluation of this type of healthcare service increases over time and with the frequency of its use, it should be developed, but targeted at groups of health problems that do not pose a serious risk of complications.

## 5. Limitations

Although we believe that our study brings an interesting input to assessing the implementation of telemedicine services from the patient’s perspective (assessing the same population year after year), we want to present some limitations within the preparation of this article. Firstly, due to the design of the survey questionnaire, we were forced to make some simplifications in assessing the frequency of use of telemedicine services. With the answers to the question on the use of telemedicine with a GP and with a specialist in the first and second year of the pandemic, we constructed a variable categorizing the frequency of telemedicine use. It must be acknowledged that this is only an approximation, as we did not ask questions about the specific number of such visits in a given year. On this basis, we categorized respondents into never, once, twice for the respective year of assessment and never, rarely, and regularly for the summative assessment. It seems, however, that the division adopted significantly reflects the rule, also emphasized in the literature, that positive perception of e-health elements increases with the frequency of their use.

## 6. Conclusions

Whether in Poland or globally, the development of digitization of the healthcare sector is constantly constrained by several barriers, which were discussed in the article. In order to fully exploit the potential of telemedicine, it is important to start with measures to minimize these barriers. Considering the differences in reported barriers to accessing telemedicine, it is imperative that any intervention to improve access has to be tailored to the specific needs of the community. Furthermore, it is recommended to follow the ideas and solutions proposed by countries where the implementation of telemedicine services is at a higher stage. It is necessary to remember that the level of patient satisfaction is a traditional quality standard, which is an important element for further prioritizing improvements in the provision of telehealth services. The functioning of telemedicine platforms, with the possibility of using images in teleconsultations, would be welcomed, with an equalization of disparities in the quality of telemedicine and DPC services between urban and rural areas, as well as between commercial and non-commercial services. This could improve the process of service provision and consequently benefit the competitiveness of telemedicine, which will be based on medical rather than organizational aspects. Nevertheless, further research is needed to identify and moderate disparities in healthcare, as well as the methods for maximizing patient and provider satisfaction with telemedicine services. This will provide insight into what further implementation of telemedicine should look like in the existing healthcare system.

## Figures and Tables

**Figure 1 ijerph-20-00115-f001:**
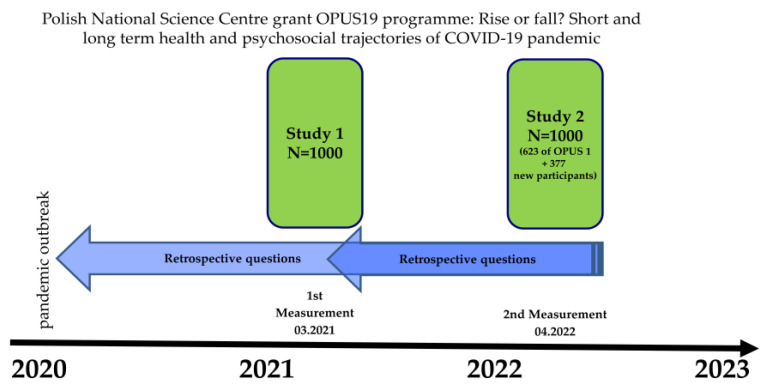
Study timeline.

**Figure 2 ijerph-20-00115-f002:**
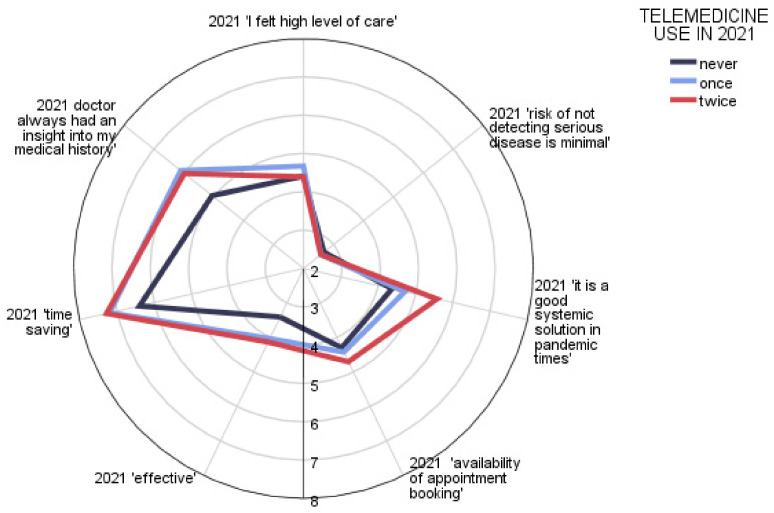
Patient opinion on telemedicine after year 1 of the pandemic.

**Figure 3 ijerph-20-00115-f003:**
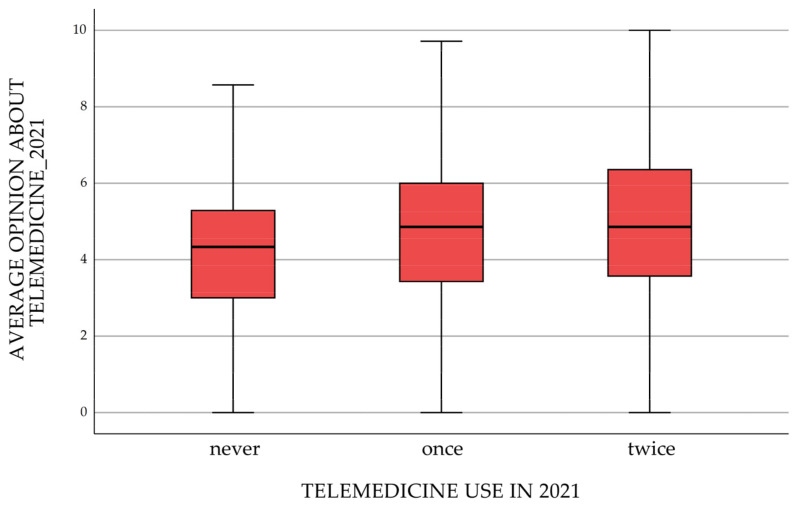
Average opinion on telemedicine after 1 year of the pandemic.

**Figure 4 ijerph-20-00115-f004:**
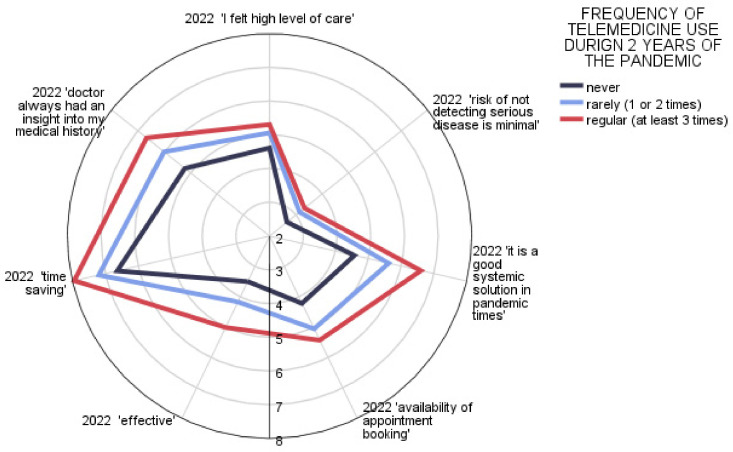
Patient opinion on telemedicine after year 2 of the pandemic (summarized).

**Figure 5 ijerph-20-00115-f005:**
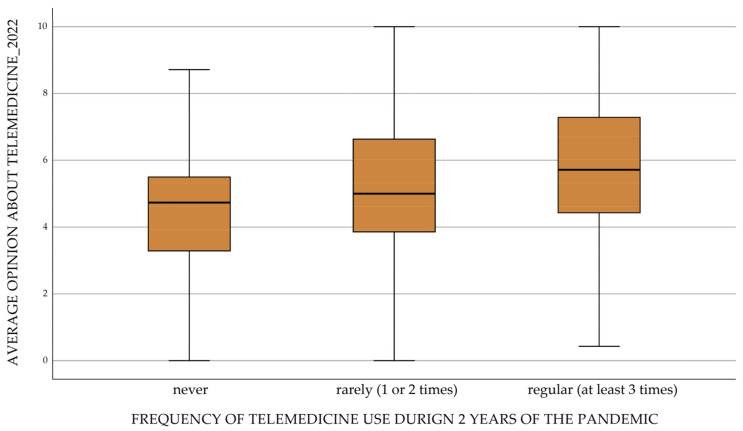
Average opinion on telemedicine after two years of the pandemic.

**Figure 6 ijerph-20-00115-f006:**
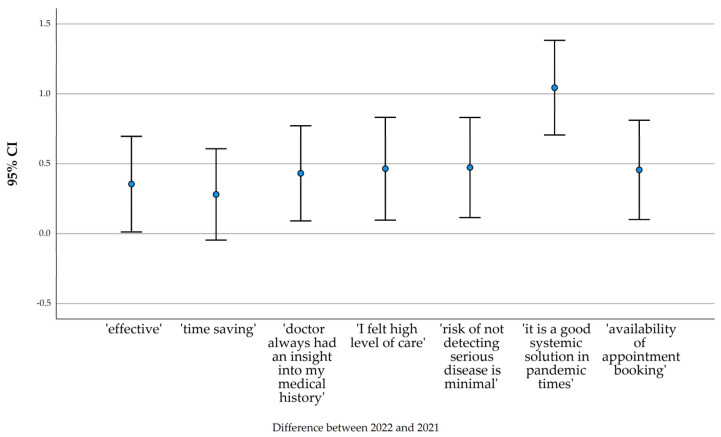
2022–2021 difference in average score per opinion item.

**Table 1 ijerph-20-00115-t001:** Characteristics of the study population.

Variables N (% of N in a Row)		Frequency of Telemedicine Use during 2 Years of the Pandemic				
		Never	Rarely (1 or 2 Times)	Regular (at Least 3 Times)	Total	*p*-Value *
		N = 136	N = 293	N = 194	N = 623	
Gender	female	58 (18.3)	153 (48.3)	106 (33.4)	317	0.082
	male	78 (25.5)	140 (45.8)	88 (28.8)	306	
Age	18–24	10 (31.3)	18 (56.3)	4 (12.5)	32	<0.001
	25–34	29 (31.5)	52 (56.5)	11 (12.0)	92	
	35–44	34 (28.8)	52 (44.1)	32 (27.1)	118	
	45–54	23 (27.4)	33 (39.3)	28 (33.3)	84	
	55–64	23 (17.8)	58 (45.0)	48 (37.2)	129	
	65+	17 (10.1)	80 (47.6)	71 (42.3)	168	
Education	lower	70 (32.0)	92 (41.4)	60 (27.0)	222	<0.001
	middle	40 (18.0)	105 (47.5)	76 (34.4)	221	
	higher	26 (14.0)	96 (53.3)	58 (32.2)	180	
City size	village/rural	56 (25.0)	112 (50.0)	56 (25.0)	224	0.042
	city/town up to 100,000	44 (21.0)	81 (39.5)	80 (39.0)	205	
	city/town of 100–500,000	22 (19.0)	60 (52.6)	32 (28.1)	114	
	city/town of more than 500,000	14 (18.0)	40 (50.0)	26 (32.5)	80	

* Pearson’s χ^2^ test.

**Table 2 ijerph-20-00115-t002:** Results of testing comparisons of telemedicine assessments after the first year of the pandemic.

Item-Opinion on Telemedicine	*p*-Value *	Pairwise Comparisons **		
		Never vs. Once	Never vs. Twice	Once vs. Twice
effective	0.063	sig.		
time saving	0.005	sig.	sig.	
insight into medical history	0.012	sig.	sig.	
high level of care	0.631			
minimal risk of not detecting disease	0.982			
good systemic solution	0.004		sig.	sig.
availability of doctor appointment	0.559	sig.	sig.	
average opinion in 2021	0.002	sig.	sig.	

* Kruskal-Wallis test; ** Mann-Whitney test with Bonferroni correction; sig. test result statistically significant after application of Bonferroni correction for multiple comparisons.

**Table 3 ijerph-20-00115-t003:** Results of testing comparisons of telemedicine assessments after two years of the pandemic (summarized).

Item-Opinion on Telemedicine	*p*-Value *	Pairwise Comparisons **		
		Never vs. Rarely	Never vs. Regular	Rarely vs. Regular
effective	<0.001	sig.	sig.	sig.
time saving	<0.001		sig.	sig.
insight into medical history	<0.001		sig.	sig.
high level of care	0.004		sig.	sig.
minimal risk of not detecting disease	0.321			
good systemic solution	0.091		sig.	
availability of doctor appointment	<0.001	sig.	sig.	sig.
average opinion in 2022	0.011	sig.	sig.	

* Kruskal-Wallis test; ** Mann-Whitney test with Bonferroni correction; sig. test result was statistically significant after application of Bonferroni correction for multiple comparisons.

**Table 4 ijerph-20-00115-t004:** Univariate and Multivariate Linear Regression of Average Opinion on telemedicine after two years of the pandemic.

Variables	Category/Values	Univariate Analysis		Multivariate Analysis (Model R^2^ = 0.082)		VIF
		β	*p*	β	*p*	
Gender	male	0.012	0.766	0.021	0.596	1.014
Age	[years]	0.167	<0.001	0.100	0.020	1.175
Education	[lower = 1 to higher = 3]	0.040	0.330	−0.016	0.707	1.121
City size	[rural = 1 to over 500 k = 4]	−0.023	0.572	−0.063	0.123	1.063
Telemedicine frequency	[never = 0 to regular = 2]	0.201	<0.001	0.159	<0.001	1.079
Vaccinated against COVID-19	[yes = 1]	−0.195	<0.001	−0.149	<0.001	1.149

β—Standardized Beta coefficient; *p*—*p*-value; VIF—Variance Inflation Factor.

## Data Availability

The dataset we generated and/or analyzed during the current study are not publicly available due to confidentiality issues but are available from the corresponding author on request.

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
