# Peer review of "Patient’s Perspective of Telemedicine in Poland—A Two-Year Pandemic Picture"

_ijerph, 2022, doi:10.3390/ijerph20010115_

Round 1
Reviewer 1 Report
Dear Authors the paper “Assessment of rapid implementation of telemedicine in Poland
from the patient's perspective - a two-year pandemic picture ‘’is really interesting, well conducted and fits the objectives of the journal; but it is necessary to review some aspects specifically i ask you to check the plagiarism of your article, I suggest you to modify the title and add the type of article.
The introduction section is very short and is needed to add other references, these article about telemedicine will help you to improve your article.
How social media meet patients’ questions: YouTubE™ review for children oral thrush PubMed ID 29460525
How social media meet patients’ questions: YouTubE™ review for mouth sores in children PubMed ID 29460528
Teledentistry in the Management of Patients with Dental and Temporomandibular Disorders Doi: https://doi.org/10.1155/2022/7091153
You need to review the grammar and English of your article
I suggest you add a table with the list of abbreviations used in the text.
Regards
Reviewer 2 Report
Dear author team,
Interesting publication you have submitted.
With much interest, I did read your manuscript and overall, I think your contribution is highly relevant.
I have concerns related to your manuscript and suggest major revisions to each subchapter, as I believe there is room for improvement to really make it a very fine contribution to the much needed data on how telemedicine can be used and sensibly be expanded in the future.
Title:
Does not match what is presented in the manuscript…you present data of one year only…
Association of researchers:
Needs to be streamlined
Abstract:
Needs to be re-written after each section is addressed and needs to better present the content of the manuscript.
Introduction:
Any explanation/context information on the polish health system is missing. The reader needs to get some basic insights to the country context where the study was conducted.
a. How is the service provision organized? Public vs private health service provision? Health insurance existing?
b. What exactly changed in the service provision due the COVID-19?
c. Who implemented these phone calls? What it the initiative of the public sector? Was this guided by any legal framework? How exactly were these phone calls/video calls done? Also for outpatient hospital consultancies with specialist?
Methods:
No mention how the questionnaire was developed, tested for understandability or how internal validity was established. In the limitation section you mention there were issues with the tool, but totally unclear how this tool was developed.
The questionnaire and its major items should be presented in a table format, for visibility and transparency purposes.
How was the CAWI technique used? Did the research team send out the online questionnaires? Was the same sample used in the follow up in 2022
In the acknowledgement you state that the study was part of an existing cohort, and in the method section you say that the sample was drawn from an opinion panel. Pls. clarify.
Results:
As the results are presented now, there is no novelty in terms of content to it. The reader does not learn anything new and what is presented is not much more than common sense, to be honest. Statistics should be used where it makes sense and not just to play around and to tell the reader the most obvious.
What is it about 2021 that you present? Data was collected in 2021 and 2022…I would expect to have the change over this one year presented in the opinion of the service users.
What is your definition of “effective” and how does it differ from “time saving”.
The narrative of the results section is less than half a page, and only contains one figure and one table. If there is not much more to analysis from the data you have, then a reconsideration of a publication is necessary.
There are some great publications out there on the topic…you may want to have a closer look at some of the really well written/analyzed manuscripts on the topic.
Discussion:
In this section you are presenting different (random?) studies from health care systems around the globe. It does not become clear to the reader what your main message is in this section and what the relevance is for your context. The section would benefit from a more systematic approach of connecting the different findings from elsewhere and argue more pointedly why they are relevant for your context or not. This section will also become more conclusive, if you strengthen the introduction chapter on the context of the polish health care system.
Acknowledgement:
It’s a bit confusing that it is stated that the study is part of a cohort…this is not explained in the methods section
References:
Just browsing the literature list, I realized some inconsistencies
Round 2
Reviewer 2 Report
Dear author team,
Thank you. In my view, your scientific contribution to the field has significantly improved with the revisions you have undertaken in your manuscript.